# Highly Sensitive MEMS Sensor Using Bimetallic Pd–Ag Nanoparticles as Catalyst for Acetylene Detection

**DOI:** 10.3390/s22197485

**Published:** 2022-10-02

**Authors:** Yuan Tian, Hui Qiao, Tao Yao, Shuguo Gao, Lujian Dai, Jun Zhao, Ying Chen, Pengcheng Xu

**Affiliations:** 1State Grid Hebei Electric Power Co., Ltd., Electric Power Research Institute, Shijiazhuang 050021, China; 2Equipment Department of State Grid Hebei Electric Power Co., Ltd., Shijiazhuang 050022, China; 3State Grid Hebei Information & Telecommunication Branch, Shijiazhuang 050013, China; 4State Key Lab of Transducer Technology, Shanghai Institute of Microsystem and Information Technology, Chinese Academy of Sciences, Shanghai 200050, China

**Keywords:** dissolved gas analysis, power transformer, fault diagnosis, acetylene sensor, MEMS, microhotplates, bimetallic catalyst, indium oxide

## Abstract

Acetylene detection plays an important role in fault diagnosis of power transformers. However, the available dissolved gas analysis (DGA) techniques have always relied on bulky instruments and are time-consuming. Herein, a high-performance acetylene sensor was fabricated on a microhotplate chip using In_2_O_3_ as the sensing material. To achieve high sensing response to acetylene, Pd–Ag core-shell nanoparticles were synthesized and used as catalysts. The transmission electron microscopy (TEM) image clearly shows that the Ag shell is deposited on one face of the cubic Pd nanoseeds. By loading the Pd–Ag bimetallic catalyst onto the surface of In_2_O_3_ sensing material, the acetylene sensor has been fabricated for acetylene detection. Due to the high catalytic performance of Pd–Ag bimetallic nanoparticles, the microhotplate sensor has a high response to acetylene gas, with a limit of detection (LOD) of 10 ppb. In addition to high sensitivity, the fabricated microhotplate sensor exhibits satisfactory selectivity, good repeatability, and fast response to acetylene. The high performance of the microhotplate sensor for acetylene gas indicates the application potential of trace acetylene detection in power transformer fault diagnosis.

## 1. Introduction

Power transformers are part of the key equipment in power plants and substations, and the reliability of their operation determines power transmission and the operation of power plants [1,2,3,4]. So far, oil-immersed power transformers have been widely used in the power industry, using transformer-specific oil for insulation and heat dissipation [4,5,6]. When an internal insulation failure occurs in the power transformer equipment, such as partial discharge caused by a short circuit in the transformer, the transformer oil will decompose to produce a series of gases, including C_2_H_2_, C_2_H_4_, CH_4_, and H_2_ [4,7,8,9]. Among them, acetylene (C_2_H_2_) is an important component in the decomposition gas of transformer oil and is often used as a characteristic gas for diagnosing some faults of transformers [10,11,12,13]. That is, the content of acetylene in the transformer oil affects the safe operation of the transformer. However, investigating the reason for acetylene production and addressing transformer failures accordingly is very difficult and time-consuming. Therefore, some gas monitoring methods are required to detect the concentration of acetylene gas in the transformer.

In recent years, dissolved gas analysis (DGA) techniques have been widely used to detect acetylene in transformer oil [4,14,15,16]. DGA techniques include oil chromatography, photoacoustic spectroscopy, infrared spectroscopy, and Raman spectroscopy [17,18,19,20,21]. However, the existing traditional DGA technologies have disadvantages, such as bulky size, high power consumption, and high cost, which limit their large-scale applications. Compared with these large-scale detection instruments, the gas sensor has advantages of small size, simple structure, and good performance, and is a new DGA technology with great application potential. A wide variety of gas sensors, including semiconductors and electrochemical, have been extensively studied for their acetylene-sensitive properties over the past few decades [8,22,23,24,25,26,27]. For example, Barsan et al. studied the sensitivity of the Ag-loaded LaFeO_3_ semiconductor to C_2_H_2_ gas, and they found that the response and selectivity of the LaFeO_3_ sensor to acetylene were optimal when the Ag loading was 0.1 wt% [28]. Lu et al. reported a mixed-potential acetylene sensor based on YSZ [29,30]. They found that composite metal oxides (perovskite, spinel, or other mixed oxides) are ideal electrode materials for mixed-potential YSZ-based gas sensors. Recently, Lu et al. reported another Ti-based composite oxide sensing electrode, which has a detection limit of 500 ppb for acetylene [29]. For on-site detection of trace gases, such as acetylene, miniature gas sensors with high sensing performance and low power consumption have attracted much attention [31,32,33,34,35,36].

In this work, we report a highly sensitive Micro-Electro-Mechanical Systems (MEMS) sensor for acetylene detection using bimetallic Pd–Ag nanoparticles as catalysts. To fabricate the MEMS sensor, we first loaded the In_2_O_3_ sensing material onto a microhotplate chip, which is batch fabricated using MEMS technology. To achieve high acetylene response, we successfully synthesized Pd–Ag bimetallic nanoparticles and used these as catalysts for acetylene detection. The as-synthesized Pd–Ag bimetallic nanoparticles significantly improve the sensing response of the In_2_O_3_-based MEMS gas sensor. Due to the use of MEMS chips, the power consumption of the fabricated sensor is only 30 mW at operating temperature. In addition to high sensitivity, the fabricated MEMS sensors are highly selective and repeatable to acetylene gas. The response speed of the sensor to acetylene was also investigated.

## 2. Materials and Methods

In_2_O_3_ NPs were synthesized using the precipitation method, which is briefly described as follows. First, 1.5 g of indium nitrate (In(NO_3_)_2_ 9H_2_O, Sigma Aldrich, Milwaukee, WI, USA) was dissolved into 20 mL of deionized (DI) water and stirred at room temperature for 30 min to form a homogeneous precursor solution. Subsequently, ammonium hydroxide solution was added dropwise to the precursor solution to adjust the pH of the solution to 10. The resulting white precipitate was centrifuged, washed three times with ethanol, and dried at 80 °C for 1 day in an air atmosphere. Finally, the product was calcined in a tube furnace at 300 °C for 2 h to obtain In_2_O_3_ NPs.

The preparation process of the Pd seeds is briefly described as follows: 100 mg of polyvinylpyrrolidone (PVP, MW ≈ 40,000), 60 mg of L-ascorbic acid (AA), and 600 mg of KBr were sequentially added into 11 mL of DI water and stirred until completely dissolved. About 55 mg of Na_2_PdCl_4_ was added to the above-mentioned solution and stirred at 80 °C for 3 h. After cooling to room temperature, the product was collected by centrifugation and washed three times with ethanol and DI water. Excess Br^−^ ions were removed by multiple centrifugations and washings. The resulting precipitate was dispersed in 8 mL of DI water for TEM characterization or as seeds for the following sample preparation.

The Pd–Ag bimetallic nanoparticles were synthesized via a modified seed-mediated growth method in the literature [37]. In order to deposit Ag shell onto the surface of the Pd seeds, 4 mL of DI water, 0.1 mL of the Pd seeds solution, 0.1 mL of PVP solution (18 mg/mL), and 0.4 mL of AA solution (65 mM) were sequentially added into a flask. The mixture was stirred at room temperature for 20 min. Meanwhile, an AgNO_3_ aqueous solution with a concentration of 1.2 mM was prepared at room temperature. The AgNO_3_ solution was injected into the flask at a constant rate with the help of a syringe pump. After the solution was injected completely, the solution was allowed to stir 2 min. After that, the product was collected by centrifugation and washed 3 times with DI water.

Microhotplates were used to fabricate the acetylene sensor with low power consumption. The microhotplates were batch fabricated using MEMS technology and the fabrication process can be found in our previous literature [38]. The In_2_O_3_ sensing material was loaded into the sensing area using a commercially available micromanipulator (Eppendorf PatchMan NP2). After sample loading, the microhotplates were aged at 400 °C for 1 week before testing. After aging, the In_2_O_3_ material has good adhesion to the microhotplate electrode, and the ohmic contact was significantly improved. In order to load catalyst onto the surface of the In_2_O_3_ sensing material, about 10 mg of Pd–Ag bimetallic nanoparticles were first dispersed into 1 mL of ethanol to form a suspension. Then, 0.1 μL of the suspension was precisely loaded onto the sensing area using a commercial inkjet printer (SonoPlot Microplotter GIX II, Middleton, WI, USA). After that, the MEMS sensor was dried in an oven at 333 K for about 2 h.

Powder X-ray diffraction (XRD) patterns were obtained by generating X-rays (40 kV, 40 mA) using a Bruker model D8 focusing diffractometer equipped with a Cu anode. The resulting X-ray beam contains Cu radiation with wavelengths λKα1 = 0.15406 nm and λKα2 = 0.15444 nm, with a Cu Kα1/Cu Kα2 intensity ratio of 2:1. Data were acquired in continuous scan mode from 20° to 70° with a sampling interval of 0.02°. The morphology of the In_2_O_3_-based sensor was characterized using a field emission scanning electron microscope (FE-SEM, model Hitachi S4800, Tokyo, Japan). The Pd and Pd–Ag catalysts were characterized using a high-resolution transmission electron microscopy (HR-TEM, FEI Tecnai G^2^ F20). The TEM was equipped with a CMOS camera (EMSIS XAROSA) and operated at 200 kV.

Standard gases, including C_2_H_2_ with desired concentrations, were purchased from Shanghai Flextronics Gas Company. All mass flow controllers (MFCs) are calibrated using digital soap film flow meters. High-purity air was used as the carrier gas, and the flow rate was set to 1000 standard cubic centimeters per minute (sccm). Using another MFC, the C_2_H_2_ flow can be further diluted with pure air to the desired concentration and then introduced into the test chamber. To evaluate the effect of humidity on the sensing response, the standard gas was flowed through water to generate a wet gas, and the value of the relative humidity (RH) of the gas flow was recorded using a digital thermo-hygrometer (Dretec, model O-251). The resistance value of the MEMS sensor was recorded in real time with a multimeter (Agilent 34401A).

## 3. Results and Discussion

### 3.1. Characterization of the Sensing Material

The as-synthesized Pd seeds were first characterized by using TEM. The TEM pattern in Figure 1a indicates that the as-synthesized sample features cube-like morphology. Figure 1a also shows that the prepared Pd nanoparticles are well-dispersed, with an average edge length of 7 nm. The high-resolution TEM (HRTEM) pattern in Figure 1b clearly shows the lattice fringes of Pd seeds. The spacing of the lattice fringes is measured as 0.19 nm, which is consistent with the (200) crystal plane of the face-centered cubic (fcc)-structured Pd. The TEM images in Figure 1a,b demonstrate that the cubic-structured Pd nanoparticles have been successfully synthesized and can be further used as seeds for the preparation of Pd–Ag bimetallic nanoparticles.

With L-ascorbic acid (AA) as the reducing agent, Ag shell layer can be controllably grown on the surface of Pd seeds to form novel Pd–Ag core-shell nanoparticles. As shown in Figure 1c,d, the Ag shell can be mainly deposited onto one of the Pd crystal planes. Based on the TEM results in Figure 1c,d, the Pd–Ag bimetallic nanoparticles with core-shell structure have been prepared successfully.

Figure 2a shows the XRD pattern of the In_2_O_3_ sample synthesized using the precipitation method. As shown in Figure 2a, the diffraction peaks located at 21.6°, 30.6°, 35.5°, 51.2°, and 60.7° can be assigned to the (112), (222), (400), (440), and (622) faces of In_2_O_3_ with cubic structure, according to the standard JCPDS card (No.: 06-0416) [39]. There is no diffraction peak that comes from the impurity that can be observed in Figure 2a. The SEM image in Figure 2b further shows that the as-synthesized In_2_O_3_ sample has a cubic morphology with a mean diameter of 1.2 μm. After In_2_O_3_ sensing material and Pd–Ag core-shell nanocatalyst have been successfully prepared, the acetylene sensor can be fabricated. In order to obtain an acetylene sensor with low power consumption, MEMS hotplate chips have been fabricated. With the aid of a microscope, the In_2_O_3_ sensing material is first loaded into the sensing area. After that, the Pd–Ag core-shell nanocatalyst can be precisely coated on the top layer of the In_2_O_3_ sensing material to form an acetylene sensor. Figure 2c shows the SEM image of a fabricated MEMS acetylene sensor, and the structure of the MEMS chip is shown in Figure 2d. The fabricated sensor uses In_2_O_3_ layer as sensing material and Pd–Ag core-shell nanoparticles as catalyst to improve the sensing performance. The resistance of the In_2_O_3_ layer is measured as sensing signal.

### 3.2. Acetylene Sensing Performance of the MEMS Sensor

In order to investigate the gas-sensing performance of the fabricated MEMS gas sensors, the operating temperature of the sensor was first optimized. In order to determine the optimized temperature, the MEMS gas sensors were exposed to 1 ppm C_2_H_2_ in the temperature range of 100 to 500 °C. As shown in Figure 3a, the response of the sensor reaches a maximum value at 400 °C. When the operating temperature is higher than 400 °C, the response drops dramatically. The reason lies in that the core-shell structure of the Pd–Ag nanoparticles will be destroyed above 400 °C and the catalytic activation will decrease accordingly. Therefore, the operating temperature for C_2_H_2_ detection using the Pd–Ag@In_2_O_3_ sensor was optimized to be 400 °C. According to the previous report, the power consumption of the MEMS sensor is measured as 28 mW at 400 °C.

In this study, the gas response is defined as R_a_/R_g_, where R_a_ and R_g_ are the sensor resistances in dry air atmosphere and during gas exposure, respectively. As shown in Figure 3b, the sensor was exposed to C_2_H_2_, with concentrations ranging from 10 ppb to 500 ppb at 400 °C, and the sensing curve was recorded in real time. Figure 3b also indicates that, as the acetylene concentration increases, the sensing response of the MEMS gas sensor increases accordingly. From the experimental results in Figure 3b, it is clear that the limit of detection (LOD) of the sensor for C_2_H_2_ is much lower than 10 ppb. It should be mentioned that, without using Pd–Ag bimetallic catalyst, the bare In_2_O_3_-based MEMS sensor has a negligible response to acetylene gas with a concentration of hundreds of ppm.

Figure 3c shows that the response of the sensor increases linearly with the increased acetylene concentration. As shown in Figure 3a, the relationship between the sensing response and the C_2_H_2_ concentration in the measured concentration range can be well fitted to a linear equation of *y* = *a* + *bx*, where *y* represents the sensing response value and *x* is the C_2_H_2_ concentration. According to the fitting results, as shown in the inset of Figure 3c, the adjusted R^2^ is as high as 99.1%.

Diagnosing faults of the transformer by detecting C_2_H_2_ requires the sensor to have good selectivity to C_2_H_2_ gas. In this work, five typical gases (or vapors), including carbon monoxide (CO), hydrogen (H_2_), methane (CH_4_), ammonia (NH_3_), and ethanol (C_2_H_5_OH), were selected as interfering gases. All interfering gases were detected at 50 ppm, while C_2_H_2_ was only 1 ppm. As shown in Figure 3d, the sensor has a high response value of 7.5 to 1 ppm C_2_H_2_, which is much higher than the response to the selected five interfering gases. The experimental results in Figure 3d show that the sensor exhibits satisfactory selectivity to C_2_H_2_, showing good application potential in fault diagnosis of power transformers.

The sensor shows good repeatability in multiple C_2_H_2_ detection experiments. To investigate repeatability, the sensor was allowed to detect 300 ppb C_2_H_2_ for three times. As shown in Figure 4a, the sensor responses to C_2_H_2_ gas are measured as 1.97, 1.99, and 1.99, respectively. The relative standard deviation (RSD) of the three response values was calculated to be 0.475% (*n* = 3), indicating that the sensor has good repeatability for C_2_H_2_ detection. The sensor exhibits a fast response speed during the C_2_H_2_ detection. Here, *t*_90_ is used to represent the response speed, which is defined as the response time from the initial state to the 90% equilibrium state [40]. Here, the sensor response to 300 ppb C_2_H_2_ is taken as an example. The *t*_90_ value is only 20 s, which indicates that the sensor responds quickly to C_2_H_2_ gas.

Since the sensor is operated at the high working temperature of 400 °C, the sensor exhibits good resistance to humidity change. As shown in Figure 4c, when the sensor detects 300 ppb of acetylene at different humidity conditions, the response values are always measured as 2, with minor fluctuations of ±0.12.

Our study further shows that the thickness of the Ag shell of the Pd–Ag catalyst has a significant effect on the sensing response, rather than the size of the Pd seeds. Here, three catalysts with a series of shell thicknesses (i.e., 1#: 4.4 nm, 2#: 8 nm, and 3#: 20 nm) were synthesized and used to detect acetylene. As shown in Figure 4d, the response of the sensor increases with the thickness of the Ag shell. By using the 3# Pd–Ag catalyst, the response of the sensor can only be improved by ~15% compared to the 2# catalyst in Figure 3. However, it is a great challenge to synthesize Pd–Ag catalysts with high Ag thickness using the method in this work, and its yields are not high. Therefore, 2# catalysts with high yields are mainly used to fabricate acetylene sensors in this work.

## 4. Conclusions

This paper presents a high-performance MEMS sensor for acetylene detection. To fabricate the sensor, the In_2_O_3_ sensing material, as well as the Pd–Ag bimetallic catalyst, were loaded onto the microhotplate chip, which is fabricated with MEMS technology. In this work, the In_2_O_3_ sensing materials were synthesized using a facile precipitation method. According to the characterization results, the prepared In_2_O_3_ sensing material has a cubic structure with an average particle size of 1.2 μm. The Pd–Ag core-shell nanoparticles were prepared by a seed synthesis method. TEM results revealed that the Ag shell was deposited onto one face of the cubic Pd nanoseeds. The fabricated MEMS sensor exhibits a good response to acetylene gas with a satisfactory LOD value of 10 ppb. The sensor also exhibits good selectivity to acetylene compared to five common interfering gases. The repeatability of the sensor was investigated, and the relative standard deviation (RSD) of three detections was only 0.475% (*n* = 3). The *t*_90_ of the sensor for 300 ppb acetylene was measured as 20 s, which indicates the fast response of the sensor to acetylene gas. In summary, the reported MEMS sensor in this work demonstrates good sensing performance to acetylene gas, which exhibits good application potentials for acetylene detection during the fault diagnosis of power transformers.

## Figures and Tables

**Figure 1 sensors-22-07485-f001:**
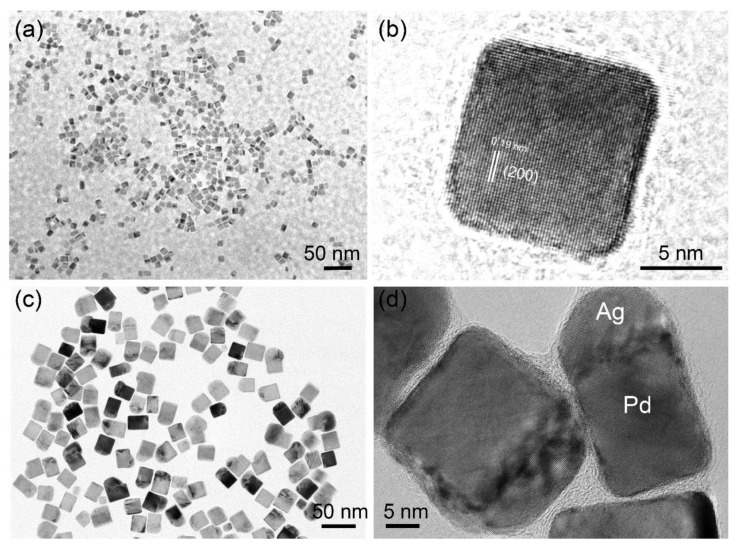
TEM images of the cubic Pd nanoparticles and Pd–Ag bimetallic catalysts. (**a**) Low-resolution and (**b**) high-resolution TEM images of Pd nanoparticles; (**c**) low-resolution TEM image of the Pd–Ag bimetallic catalysts; (**d**) high-resolution TEM image clearly shows that the Ag shell was deposited onto one face of cubic Pd nanoparticles.

**Figure 2 sensors-22-07485-f002:**
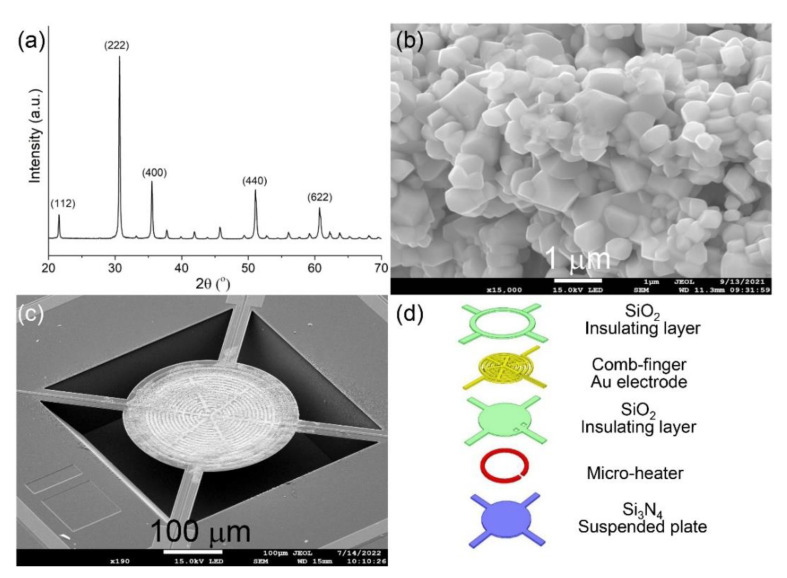
(**a**) XRD pattern and (**b**) SEM image of the prepared In_2_O_3_ sensing material; (**c**) SEM image and (**d**) structure of the fabricated MEMS sensor.

**Figure 3 sensors-22-07485-f003:**
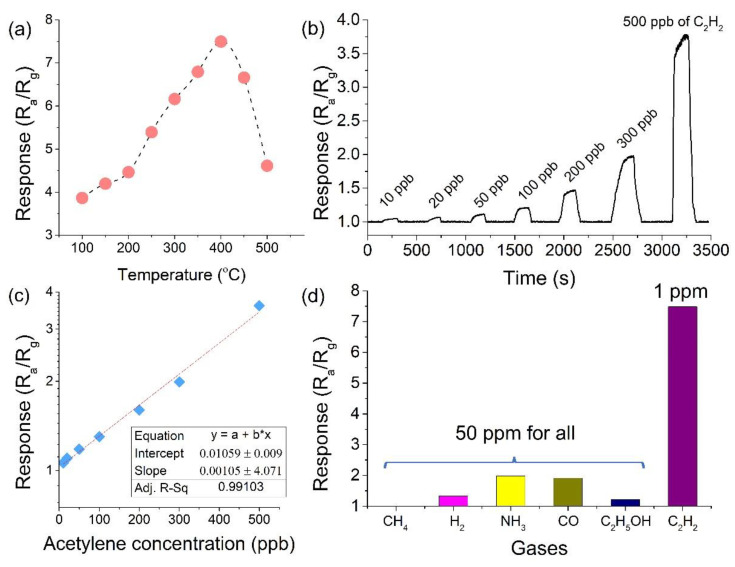
Sensing response of the fabricated MEMS sensor. (**a**) Responses of the sensor measured at various working temperatures; (**b**) real-time recorded sensing curve; (**c**) linear relationship between the sensing response and acetylene concentration; (**d**) cross-responses of the sensor to interfering gases.

**Figure 4 sensors-22-07485-f004:**
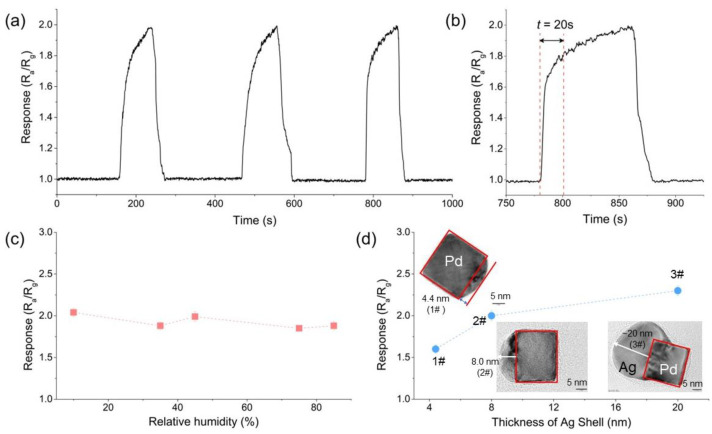
(**a**) Repeatability of the sensor to 300 ppb acetylene; (**b**) response speed of the sensor to 300 ppb acetylene; (**c**) influence of humidity change on the sensing response; (**d**) influence of the thickness of Ag shell on the sensing response. In (**c**,**d**), the concentration of acetylene is 300 ppb.

## Data Availability

Not applicable.

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
