# Peer review of "Highly Sensitive MEMS Sensor Using Bimetallic Pd–Ag Nanoparticles as Catalyst for Acetylene Detection"

_sensors, 2022, doi:10.3390/s22197485_

Round 1

Reviewer 1 Report

This manuscript deals with a C2H2 sensor using In2O3 and Pd-Ag nanoparticles on a MEMS-based microhotplate structure. The authors demonstrated ppb-order sensitive C2H2 sensing and selectivity. But there are the following uncertain points. To further consider this manuscript for publication, these points should be fully addressed.

1.      It is unclear that the mechanism of selective adsorption to C2H2. Which material is the C2H2 receptor? Also, which material’s resistance is measured for the resistance change during gas adsorption?

2.      In2O3 was loaded on the microhotplate using a manipulator. How was the electrode adhered to the freestanding membrane? Does the loaded In2O3 by postprocessing have an ohmic contact with the electrode on the membrane?

3.      Could you consider why the resistance change is largest at 400 °C.

4.      If the concentration dependence of Fig.3(c) is created from the resistance ratio of Fig. 3(b), is it correct that the vertical axis c is 0? Also, the resistance change seems to saturate at 100 ppb C2H2, but the other responses do not. Please explain these differences. Also, can the authors correctly read the concentration dependence from the non-saturated response?

5.      What is the humidity percentage during the gas measurement? Please comment on the humidity dependence of the gas measurement.

Author Response

Our responses to Reviewer 1 can be found in the attached file.

Reviewer 2 Report

This is a good technical article showing the high performance MEMS sensor for detecting acetylene. The high performance of this sensor indicates its potential for the detection of very small amounts of acetylene in the diagnosis of power transformer failures.

The manuscript is well written and nicely presented, with a balance between descriptive text and discourse, and with a practical illustration of the methods presented. The methodology used is appropriate.

Author Response

Thank you very much for your positive comments!

Reviewer 3 Report

The authors reported a highly sensitive MEMS sensor for acetylene detection. They used Pd-Ag core-shell nanoparticles as catalysts and found that Pd-Ag bimetallic nanoparticles significantly improve the sensing response of the sensor. The findings are interesting and useful. My concerns are:

1. Does particle size affect the results? If yes, what is effect?

2. Dose particle distribution affect the conclusion? If yes, what is the effect?

3. Why does the response depend linearly on the concentration rather than the Langmuir relation?

4. Does humidity affect the response?

Author Response

Our responses to the Reviewer 3 can be found in the attached file.

Round 2

Reviewer 3 Report

The authors addressed most of my concerns. I have no further objections.